# A Non-convex One-Pass Framework for Generalized Factorization Machine and Rank-One Matrix Sensing

**Ming Lin**
University of Michigan
linmin@umich.edu

**Jieping Ye**
University of Michigan
jpye@umich.edu

## Abstract

We develop an efficient alternating framework for learning a generalized version of Factorization Machine (gFM) on steaming data with provable guarantees. When the instances are sampled from $d$ dimensional random Gaussian vectors and the target second order coefficient matrix in gFM is of rank $k$, our algorithm converges linearly, achieves $O(\epsilon)$ recovery error after retrieving $O(k^3 d \log(1/\epsilon))$ training instances, consumes $O(kd)$ memory in one-pass of dataset and only requires matrix-vector product operations in each iteration. The key ingredient of our framework is a construction of an estimation sequence endowed with a so-called Conditionally Independent RIP condition (CI-RIP). As special cases of gFM, our framework can be applied to symmetric or asymmetric rank-one matrix sensing problems, such as inductive matrix completion and phase retrieval.

## 1   Introduction

Linear models are one of the foundations of modern machine learning due to their strong learning guarantees and efficient solvers [Koltchinskii, 2011]. Conventionally linear models only consider the first order information of the input feature which limits their capacity in non-linear problems. Among various efforts extending linear models to the non-linear domain, the Factorization Machine [Rendle, 2010] (FM) captures the second order information by modeling the pairwise feature interaction in regression under low-rank constraints. FMs have been found successful in many applications, such as recommendation systems [Rendle et al., 2011] and text retrieval [Hong et al., 2013]. In this paper, we consider a generalized version of FM called gFM which removes several redundant constraints in the original FM such as positive semi-definite and zero-diagonal, leading to a more general model without sacrificing its learning ability. From theoretical side, the gFM includes rank-one matrix sensing [Zhong et al., 2015, Chen et al., 2015, Cai and Zhang, 2015, Kueng et al., 2014] as a special case, where the latter one has been studied widely in context such as inductive matrix completion [Jain and Dhillon, 2013] and phase retrieval [Candes et al., 2011].

Despite of the popularity of FMs in industry, there is rare theoretical study of learning guarantees for FMs. One of the main challenges in developing a provable FM algorithm is to handle its symmetric rank-one matrix sensing operator. For conventional matrix sensing problems where the matrix sensing operator is RIP, there are several alternating methods with provable guarantees [Hardt, 2013, Jain et al., 2013, Hardt and Wootters, 2014, Zhao et al., 2015a,b]. However, for a symmetric rank-one matrix sensing operator, the RIP condition doesn't hold trivially which turns out to be the main difficulty in designing efficient provable FM solvers.

In rank-one matrix sensing, when the sensing operator is asymmetric, the problem is also known as inductive matrix completion which can be solved via alternating minimization with a global linear convergence rate [Jain and Dhillon, 2013, Zhong et al., 2015]. For symmetric rank-one matrix sensing operators, we are not aware of any efficient solver by the time of writing this paper. In a special case when the target matrix is of rank one, the problem is called "phase retrieval" whose convex solver

is first proposed by Candes et al. [2011] then alternating methods are provided in [Lee et al., 2013, Netrapalli et al., 2013]. While the target matrix is of rank $k > 1$ , only convex methods minimizing the trace norm have been proposed recently, which are computationally expensive [Kueng et al., 2014, Cai and Zhang, 2015, Chen et al., 2015, Davenport and Romberg, 2016].

Despite of the above fundamental challenges, extending rank-one matrix sensing algorithm to gFM itself is difficult. Please refer to Section 2.1 for an in-depth discussion. The main difficulty is due to the first order term in the gFM formulation, which cannot be trivially converted to a standard matrix sensing problem.

In this paper, we develop a unified theoretical framework and an efficient solver for generalized Factorization Machine and its special cases such as rank-one matrix sensing, either symmetric or asymmetric. The key ingredient is to show that the sensing operator in gFM satisfies a so-called Conditionally Independent RIP condition (CI-RIP, see Definition 2) . Then we can construct an estimation sequence via noisy power iteration [Hardt and Price, 2013]. Unlike previous approaches, our method does not require alternating minimization or choosing the step-size as in alternating gradient descent. The proposed method works on steaming data, converges linearly and has $O(kd)$ space complexity for a $d$-dimension rank-$k$ gFM model. The solver achieves $O(\epsilon)$ recovery error after retrieving $O(k^3 d \log(1/\epsilon))$ training instances.

The remainder of this paper is organized as following. In Section 2, we introduce necessary notation and background of gFM. Subsection 2.1 investigates several fundamental challenges in depth. Section 3 presents our algorithm, called One-Pass gFM, followed by its theoretical guarantees. Our analysis framework is presented in Section 4. Section 5 concludes this paper.

## 2    Generalized Factorization Machine (gFM)

In this section, we first introduce necessary notation and background of FM and its generalized version gFM. Then in Subsection 2.1, we reveal the connection between gFM and rank-one matrix sensing followed by several fundamental challenges encountered when applying frameworks of rank-one matrix sensing to gFM.

The FM predicts the labels of instances by not only their features but also high order interactions between features. In the following, we focus on the second order FM due to its popularity. Suppose we are given $N$ training instances $\boldsymbol{x}_i \in \mathbb{R}^d$ independently and identically (I.I.D.) sampled from the standard Gaussian distribution and so are their associated labels $y_i \in \mathbb{R}$ . Denote the feature matrix $X = [\boldsymbol{x}_1, \boldsymbol{x}_2, \cdots, \boldsymbol{x}_n] \in \mathbb{R}^{d \times n}$ and the label vector $\boldsymbol{y} = [y_1, y_2, \cdots, y_n]^\top \in \mathbb{R}^n$ . In second order FM, $y_i$ is assumed to be generated from a target vector $\boldsymbol{w}^* \in \mathbb{R}^d$ and a target rank $k$ matrix $M^* \in \mathbb{R}^{d \times d}$ satisfying

$$y_i = \boldsymbol{x}_i^\top \boldsymbol{w}^* + \boldsymbol{x}_i^\top M^* \boldsymbol{x}_i + \xi_i \tag{1}$$

where $\xi_i$ is a random subgaussian noise with proxy variance $\xi^2$ . It is often more convenient to write Eq. (1) in matrix form. Denote the linear operator $\mathcal{A} : \mathbb{R}^{d \times d} \to \mathbb{R}^n$ as $\mathcal{A}(M) \triangleq [\langle A_1, M \rangle, \langle A_2, M \rangle, \cdots, \langle A_n, M \rangle]^\top$ where $A_i = \boldsymbol{x}_i \boldsymbol{x}_i^\top$ . Then Eq. (1) has a compact form:

$$\boldsymbol{y} = X^\top \boldsymbol{w}^* + \mathcal{A}(M^*) + \boldsymbol{\xi} . \tag{2}$$

The FM model given by Eq. (2) consists of two components: the first order component $X^\top \boldsymbol{w}^*$ and the second order component $\mathcal{A}(M^*)$ . The component $\mathcal{A}(M^*)$ is a symmetric rank-one Gaussian measurement since $\mathcal{A}_i(M) = \boldsymbol{x}_i^\top M \boldsymbol{x}_i$ where the left/right design vectors ($\boldsymbol{x}_i$ and $\boldsymbol{x}_i^\top$) are identical. The original FM requires that $M^*$ should be positive semi-definite and the diagonal elements of $M^*$ should be zero. However our analysis shows that both constraints are redundant for learning Eq. 2. Therefore in this paper we consider a generalized version of FM which we call gFM where $M^*$ is only required to be symmetric and low rank. To make the recovery of $M^*$ well defined, it is necessary to assume $M^*$ to be symmetric. Indeed for any asymmetric matrix $M^*$, there is always a symmetric matrix $M^*_{\text{sym}} = (M^* + M^{*\top})/2$ such that $\mathcal{A}(M^*) = \mathcal{A}(M^*_{\text{sym}})$ thus the symmetric constraint does not affect the model. Another standard assumption in rank-one matrix sensing is that the rank of $M^*$ should be no more than $k$ for $k \ll d$. When $\boldsymbol{w}^* = 0$, gFM is equal to the symmetric rank-one matrix sensing problem. Recent researches have proposed several convex programming methods based on the trace norm minimization to recover $M^*$ with a sampling complexity on order of $O(k^3 d)$ [Candes

et al., 2011, Cai and Zhang, 2015, Kueng et al., 2014, Chen et al., 2015, Zhong et al., 2015]. Some authors also call gFM as second order polynomial network [Blondel et al., 2016].

When $d$ is much larger than $k$, the convex programming on the trace norm or nuclear norm of $M^*$ becomes difficult since $M^*$ can be a $d \times d$ dense matrix. Although modern convex solvers can scale to large $d$ with reasonable computational cost, a more popular strategy to efficiently estimate $\boldsymbol{w}^*$ and $M^*$ is to decompose $M^*$ as $UV^\top$ for some $U, V \in \mathbb{R}^{d \times k}$, then alternatively update $\boldsymbol{w}, U, V$ to minimize the empirical loss function

$$\min_{\boldsymbol{w}, U, V} \frac{1}{2N} \|\boldsymbol{y} - X^\top \boldsymbol{w} - \mathcal{A}(UV^\top)\|_2^2 . \tag{3}$$

The loss function in Eq. (3) is non-convex. It is even unclear whether an estimator of the optimal solution $\{\boldsymbol{w}^*, M^*\}$ of Eq. (3) with a polynomial time complexity exists or not.

In our analysis, we denote $M + O(\epsilon)$ as a matrix $M$ plus a perturbation matrix whose spectral norm is bounded by $\epsilon$. We use $\|\cdot\|_2$, $\|\cdot\|_F$, $\|\cdot\|_*$ to denote the matrix spectral norm, Frobenius norm and nuclear norm respectively. To abbreviate the high probability bound, we denote $C = \text{polylog}(d, n, T, 1/\eta)$ to be a constant polynomial logarithmic in $\{d, n, T, 1/\eta\}$. The eigenvalue decomposition of $M^*$ is $M^* = U^* \Lambda^* U^{*\top}$ where $U^* \in \mathbb{R}^{d \times k}$ is the top-$k$ eigenvectors of $M^*$ and $\Lambda^* = \text{diag}(\lambda_1^*, \lambda_2^*, \cdots, \lambda_k^*)$ are the corresponding eigenvalues sorted by $|\lambda_i| \geq |\lambda_{i+1}|$. Let $\sigma_i^* = |\lambda_i^*|$ denote the singular value of $M^*$ and $\sigma_i\{M\}$ be the $i$-th largest singular value of $M$. $U_\perp^*$ denotes an matrix whose columns are the orthogonal basis of the complementary subspace of $U^*$.

## 2.1 gFM and Rank-One Matrix Sensing

When $\boldsymbol{w}^* = 0$ in Eq. (1), the gFM becomes the symmetric rank-one matrix sensing problem. While the recovery ability of rank-one matrix sensing is somehow provable recently despite of the computational issue, it is not the case for gFM. It is therefore important to discuss the differences between gFM and rank-one matrix sensing to give us a better understanding of the fundamental barriers in developing provable gFM algorithm.

In the rank-one matrix sensing problem, a relaxed setting is to assume that the sensing operator is asymmetric, which is defined by $\mathcal{A}_i^{\text{asy}}(M) = \boldsymbol{u}_i^\top M \boldsymbol{v}_i$ where $\boldsymbol{u}_i$ and $\boldsymbol{v}_i$ are independent random vectors. Under this setting, the recovery ability of alternating methods is provable [Jain and Dhillon, 2013]. However, existing analyses cannot be generalized to their symmetric counterpart, since $\boldsymbol{u}_i$ and $\boldsymbol{v}_i$ are not allowed to be dependent in these frameworks. For example, the sensing operator $\mathcal{A}^{\text{asy}}(\cdot)$ is unbiased ( $E\mathcal{A}^{\text{asy}}(\cdot) = 0$) but the symmetric sensing operator is clearly not [Cai and Zhang, 2015]. Therefore, the asymmetric setting oversimplifies the problem and loses important structure information which is critical to gFM.

As for the symmetric rank-one matrix sensing operator, the state-of-the-art estimator is based on the trace norm convex optimization [Tropp, 2014, Chen et al., 2015, Cai and Zhang, 2015], which is computationally expensive. When $\boldsymbol{w}^* \neq \boldsymbol{0}$, the gFM has an extra perturbation term $X^\top \boldsymbol{w}^*$. This first order perturbation term turns out to be a fundamental challenge in theoretical analysis. One might attempt to merge $\boldsymbol{w}^*$ into $M^*$ in order to convert gFM as a rank $(k+1)$ matrix sensing problem. For example, one may extend the feature $\hat{\boldsymbol{x}}_i \triangleq [\boldsymbol{x}_i, 1]^\top$ and the matrix $\hat{M}^* = [M^*; \boldsymbol{w}^{*\top}] \in \mathbb{R}^{(d+1) \times d}$. However, after this simple extension, the sensing operator becomes $\hat{\mathcal{A}}(M^*) = \hat{\boldsymbol{x}}_i^\top \hat{M}^* \boldsymbol{x}_i$. It is no longer symmetric. The left/right design vector is neither independent nor identical. Especially, not all dimensions of $\hat{\boldsymbol{x}}_i$ are random variables. According to the above discussion, the conditions to guarantee the success of rank-one matrix sensing do not hold after feature extension and all the mentioned analyses cannot be directly applied.

## 3 One-Pass gFM

In this section, we present the proposed algorithm, called One-Pass gFM followed by its theoretical guarantees. We will focus on the intuition of our algorithm. A rigorous theoretical analysis is presented in the next section.

The One-Pass gFM is a mini-batch algorithm. In each mini-batch, it processes $n$ training instances and then alternatively updates parameters. The iteration will continue until $T$ mini-batch updates.

---

**Algorithm 1** One-Pass gFM

---

**Require:** The mini-batch size $n$, number of total mini-batch update $T$, training instances $X = [\boldsymbol{x}_1, \boldsymbol{x}_2, \cdots \boldsymbol{x}_{nT}]$, $\boldsymbol{y} = [y_1, y_2, \cdots, y_{nT}]^\top$, desired rank $k \geq 1$.

**Ensure:** $\boldsymbol{w}^{(T)}, U^{(T)}, V^{(T)}$.

1: Define $M^{(t)} \triangleq (U^{(t)} V^{(t)\top} + V^{(t)} U^{(t)\top})/2$, $H_1^{(t)} \triangleq \frac{1}{2n} \mathcal{A}'(\boldsymbol{y} - \mathcal{A}(M^{(t)}) - X^{(t)\top} \boldsymbol{w}^{(t)})$, $h_2^{(t)} \triangleq \frac{1}{n} \mathbf{1}^\top (\boldsymbol{y} - \mathcal{A}(M^{(t)}) - X^{(t)\top} \boldsymbol{w}^{(t)})$, $\boldsymbol{h}_3^{(t)} \triangleq \frac{1}{n} X^{(t)} (\boldsymbol{y} - \mathcal{A}(M^{(t)}) - X^{(t)\top} \boldsymbol{w}^{(t)})$.

2: Initialize: $\boldsymbol{w}^{(0)} = \mathbf{0}$, $V^{(0)} = 0$. $U^{(0)} = \text{SVD}(H_1^{(0)} - \frac{1}{2} h_2^{(0)} I, k)$, that is, the top-$k$ left singular vectors.

3: **for** $t = 1, 2, \cdots, T$ **do**

4:  Retrieve $n$ training instances $X^{(t)} = [\boldsymbol{x}_{(t-1)n+1}, \cdots, \boldsymbol{x}_{(t-1)n+n}]$. Define $\mathcal{A}(M) \triangleq [X_i^{(t)\top} M X_i^{(t)}]_{i=1}^n$.

5:  $\hat{U}^{(t)} = (H_1^{(t-1)} - \frac{1}{2} h_2^{(t-1)} I + M^{(t-1)\top}) U^{(t-1)}$.

6:  Orthogonalize $\hat{U}^{(t)}$ via QR decomposition: $U^{(t)} = \text{QR}\left(\hat{U}^{(t)}\right)$.

7:  $\boldsymbol{w}^{(t)} = \boldsymbol{h}_3^{(t-1)} + \boldsymbol{w}^{(t-1)}$.

8:  $V^{(t)} = (H_1^{(t-1)} - \frac{1}{2} h_2^{(t-1)} I + M^{(t-1)}) U^{(t)}$

9: **end for**

10: **Output:** $\boldsymbol{w}^{(T)}, U^{(T)}, V^{(T)}$.

---

Since gFM deals with a non-convex learning problem, the conventional gradient descent framework hardly works to show the global convergence. Instead, our method is based on a construction of an estimation sequence. Intuitively, when $\boldsymbol{w}^* = \mathbf{0}$, we will show in the next section that $\frac{1}{n} \mathcal{A}' \mathcal{A}(M) \approx 2M + \text{tr}(M) I$ and $\text{tr}(M) \approx \frac{1}{n} \mathbf{1}^\top \mathcal{A}(M)$. Since $\boldsymbol{y} \approx \mathcal{A}(M^*)$, we can estimate $M^*$ via $\frac{1}{2n} \mathcal{A}'(\boldsymbol{y}) - \frac{1}{n} \mathbf{1}^\top \boldsymbol{y} I$. But this simple construction cannot generate a convergent estimation sequence since the perturbation terms in the above approximate equalities cannot be reduced along iterations. To overcome this problem, we replace $\mathcal{A}(M^*)$ with $\mathcal{A}(M^* - M^{(t)})$ in our construction. Then the perturbation terms will be on order of $O(\|M^* - M^{(t)}\|_2)$. When $\boldsymbol{w}^* \neq \mathbf{0}$, we can apply a similar trick to construct its estimation sequence via the second and the third order moments of $X$. Algorithm 1 gives a step-by-step description of our algorithm[1].

In Algorithm 1, we only need to store $\boldsymbol{w}^{(t)} \in \mathbb{R}^d$, $U^{(t)}, V^{(t)} \in \mathbb{R}^{d \times k}$. Therefore the space complexity is $O(d + kd)$. The auxiliary variables $M^{(t)}, H_1^{(t)}, h_2^{(t)}, \boldsymbol{h}_3^{(t)}$ can be implicitly presented by $\boldsymbol{w}^{(t)}, U^{(t)}, V^{(t)}$. In each mini-batch updating, we only need matrix-vector product operations which can be efficiently implemented on many computation architectures. We use truncated SVD to initialize gFM, a standard initialization step in matrix sensing. We do not require this step to be computed exactly but up to an accuracy of $O(\delta)$ where $\delta$ is the RIP constant. The QR step on line 6 requires $O(k^2 d)$ operations. Compared with SVD which requires $O(kd^2)$ operations, the QR step is much more efficient when $d \gg k$. Algorithm 1 retrieves instances streamingly, a favorable behavior on systems with high speed cache. Finally, we export $\boldsymbol{w}^{(T)}, U^{(T)}, V^{(T)}$ as our estimation of $\boldsymbol{w}^* \approx \boldsymbol{w}^{(T)}$ and $M^* \approx U^{(T)} V^{(T)\top}$.

Our main theoretical result is presented in the following theorem, which gives the convergence rate of recovery and sampling complexity of gFM when $M^*$ is low rank and the noise $\boldsymbol{\xi} = \mathbf{0}$.

**Theorem 1.** *Suppose $\boldsymbol{x}_i$'s are independently sampled from the standard Gaussian distribution. $M^*$ is a rank $k$ matrix. The noise $\boldsymbol{\xi} = \mathbf{0}$. Then with a probability at least $1 - \eta$, there exists a constant $C$ and a constant $\delta < 1$ such that*

$$\|\boldsymbol{w}^* - \boldsymbol{w}^{(t)}\|_2 + \|M^* - M^{(t)}\|_2 \leq \delta^t (\|\boldsymbol{w}^*\|_2 + \|M^*\|_2)$$

*provided $n \geq C(4\sqrt{5}\sigma_1^*/\sigma_k^* + 3)^2 k^3 d/\delta^2$, $\delta \leq \frac{(4\sqrt{5}\sigma_1^*/\sigma_k^* + 3)\sigma_k^*}{4\sqrt{5}\sigma_1^* + 3\sigma_k^* + 4\sqrt{5}\|\boldsymbol{w}^*\|_2^2}$.*

Theorem 1 shows that $\{\boldsymbol{w}^{(t)}, M^{(t)}\}$ will converge to $\{\boldsymbol{w}^*, M^*\}$ linearly. The convergence rate is controlled by $\delta$, whose value is on order of $O(1/\sqrt{n})$. A small $\delta$ will result in a fast convergence rate

but a large sampling complexity. To reduce the sampling complexity, a large $\delta$ is preferred. The largest allowed $\delta$ is bounded by $O(1/(\|M^*\|_2 + \|\boldsymbol{w}^*\|_2))$. The sampling complexity is $O((\sigma_1^*/\sigma_k^*)^2 k^3 d)$. If $M^*$ is not well conditioned, it is possible to remove $(\sigma_1^*/\sigma_k^*)^2$ in the sampling complexity by a procedure called "soft-deflation" [Jain et al., 2013, Hardt and Wootters, 2014]. By theorem 1, gFM achieves $\epsilon$ recovery error after retrieving $nT = O(k^3 d \log((\|\boldsymbol{w}^*\|_2 + \|M^*\|_2)/\epsilon))$ instances.

The noisy case where $M^*$ is not exactly low rank and $\xi > 0$ is more intricate therefore we postpone it to Subsection 4.1. The main conclusion is similar to the noise-free case Theorem 1 under a small noise assumption.

## 4 Theoretical Analysis

In this section, we give the sketch of our proof of Theorem 1. Omitted details are postponed to appendix.

From high level, our proof constructs an estimation sequence $\{\widetilde{\boldsymbol{w}}^{(t)}, \widetilde{M}^{(t)}, \epsilon_t\}$ such that $\epsilon_t \to 0$ and $\|\boldsymbol{w}^* - \widetilde{\boldsymbol{w}}^{(t)}\|_2 + \|M^* - \widetilde{M}^{(t)}\|_2 \le \epsilon_t$. In conventional matrix sensing, this construction is possible when the sensing matrix satisfies the Restricted Isometric Property (RIP) [Candès and Recht, 2009]:

**Definition 2** ($\ell_2$-norm RIP). A sensing operator $\mathcal{A}$ is $\ell_2$-norm $\delta_k$-RIP if for any rank $k$ matrix $M$,

$$(1 - \delta_k)\|M\|_F^2 \le \frac{1}{n}\|\mathcal{A}(M)\|_2^2 \le (1 + \delta_k)\|M\|_F^2 .$$

When $\mathcal{A}$ is $\ell_2$-norm $\delta_k$-RIP for any rank $k$ matrix $M$, $\mathcal{A}'\mathcal{A}$ is nearly isometric [Jain et al., 2012], which implies $\|M - \mathcal{A}'\mathcal{A}(M)/n\|_2 \le \delta$. Then we can construct our estimation sequence as following:

$$\widetilde{M}^{(t)} = \frac{1}{n}\mathcal{A}'\mathcal{A}(M^* - \widetilde{M}^{(t-1)}) + \widetilde{M}^{(t-1)} , \quad \widetilde{\boldsymbol{w}}^{(t)} = (I - \frac{1}{n}XX^\top)(\boldsymbol{w}^* - \widetilde{\boldsymbol{w}}^{(t-1)}) + \widetilde{\boldsymbol{w}}^{(t-1)} .$$

However, in gFM and symmetric rank-one matrix sensing, the $\ell_2$-norm RIP condition cannot be satisfied with high probability [Cai and Zhang, 2015]. To establish an RIP-like condition for rank-one matrix sensing, several variants have been proposed, such as the $\ell_2/\ell_1$-RIP condition [Cai and Zhang, 2015, Chen et al., 2015]. The essential idea of these variants is to replace the $\ell_2$-norm $\|\mathcal{A}(M)\|_2$ with $\ell_1$-norm $\|\mathcal{A}(M)\|_1$ then a similar norm inequality can be established for all low rank matrix again. However, even using these $\ell_1$-norm RIP variants, we are still unable to design an efficient alternating algorithm. All these $\ell_1$-norm RIP variants have to deal with trace norm programming problems. In fact, it is impossible to construct an estimation sequence based on $\ell_1$-norm RIP because we require $\ell_2$-norm bound on $\mathcal{A}'\mathcal{A}$ during the construction.

A key ingredient of our framework is to propose a novel $\ell_2$-norm RIP condition to overcome the above difficulty. The main technique reason for the failure of conventional $\ell_2$-norm RIP is that it tries to bound $\mathcal{A}'\mathcal{A}(M)$ over all rank $k$ matrices. This is too aggressive to be successful in rank-one matrix sensing. Regarding to our estimation sequence, what we really need is to make the RIP hold for current low rank matrix $M^{(t)}$. Once we update our estimation $M^{(t+1)}$, we can regenerate a new sensing operator independent of $M^{(t)}$ to avoid bounding $\mathcal{A}'\mathcal{A}$ over all rank $k$ matrices. To this end, we propose the Conditionally Independent RIP (CI-RIP) condition.

**Definition 3** (CI-RIP). A matrix sensing operator $\mathcal{A}$ is Conditionally Independent RIP with constant $\delta_k$, if for a fixed rank $k$ matrix $M$, $\mathcal{A}$ is sampled independently regarding to $M$ and satisfies

$$\|(I - \frac{1}{n}\mathcal{A}'\mathcal{A})M\|_2^2 \le \delta_k . \tag{4}$$

An $\ell_2$-norm or $\ell_1$-norm RIP sensing operator is naturally CI-RIP but the reverse is not true. In CI-RIP, $\mathcal{A}$ is no longer a fixed but random sensing operator independent of $M$. In one-pass algorithm, this is achievable if we always retrieve new instances to construct $\mathcal{A}$ in one mini-batch updating. Usually Eq. (4) doesn't hold in a batch method since $M^{(t+1)}$ depends on $\mathcal{A}(M^{(t)})$.

An asymmetric rank-one matrix sensing operator is clearly CI-RIP due to the independency between left/right design vectors. But a symmetric rank-one matrix sensing operator is not CI-RIP. In fact it is a biased estimator since $E(\boldsymbol{x}^\top M \boldsymbol{x}) = \text{tr}(M)$. To this end, we propose a shifted version of CI-RIP for symmetric rank-one matrix sensing operator in the following theorem. This theorem is the key tool in our analysis.

**Theorem 4** (Shifted CI-RIP). *Suppose $\boldsymbol{x}_i$ are independent standard random Gaussian vectors, $M$ is a fixed symmetric rank $k$ matrix independent of $\boldsymbol{x}_i$ and $\boldsymbol{w}$ is a fixed vector. Then with a probability at least $1 - \eta$, provided $n \geq Ck^3d/\delta^2$,*

$$\|\frac{1}{2n}\mathcal{A}'\mathcal{A}(M) - \frac{1}{2}\mathrm{tr}(M)I - M\|_2 \leq \delta\|M\|_2 .$$

Theorem 4 shows that $\frac{1}{2n}\mathcal{A}'\mathcal{A}(M)$ is nearly isometric after shifting by its expectation $\frac{1}{2}\mathrm{tr}(M)I$. The RIP constant $\delta = O(\sqrt{k^3d/n})$. In gFM, we choose $M = M^* - M^{(t)}$ therefore $M$ is of rank $3k$.

Under the same settings of Theorem 4, suppose that $d \geq C$ then the following lemmas hold true with a probability at least $1 - \eta$ for fixed $\boldsymbol{w}$ and $M$.

**Lemma 5.** $|\frac{1}{n}\mathbf{1}^\top\mathcal{A}(M)) - \mathrm{tr}(M)| \leq \delta\|M\|_2$ *provided* $n \geq Ck/\delta^2$.

**Lemma 6.** $|\frac{1}{n}\mathbf{1}^\top X^\top\boldsymbol{w}| \leq \|\boldsymbol{w}\|_2\delta$ *provided* $n \geq C/\delta^2$.

**Lemma 7.** $\|\frac{1}{n}\mathcal{A}'(X^\top\boldsymbol{w})\|_2 \leq \|\boldsymbol{w}\|_2\delta$ *provided* $n \geq Cd/\delta^2$.

**Lemma 8.** $\|\frac{1}{n}X^\top\mathcal{A}(M)\|_2 \leq \|M\|_2\delta$ *provided* $n \geq Ck^2d/\delta^2$.

**Lemma 9.** $\|I - \frac{1}{n}XX^\top\|_2 \leq \delta$ *provided* $n \geq Cd/\delta^2$.

Equipping with the above lemmas, we construct our estimation sequence as following.

**Lemma 10.** *Let $M^{(t)}, H_1^{(t)}, h_2^{(t)}, \boldsymbol{h}_3^{(t)}$ be defined as in Algorithm 1. Define $\epsilon_t = \|\boldsymbol{w}^* - \boldsymbol{w}^{(t)}\|_2 + \|M^* - M^{(t)}\|_2$. Then with a probability at least $1 - \eta$, provided $n \geq Ck^3d/\delta^2$,*

$$H_1^{(t)} = M^* - M^{(t)} + \mathrm{tr}(M^* - M^{(t)})I + O(\delta\epsilon_t), \ h_2^{(t)} = \mathrm{tr}(M^* - M^{(t)}) + O(\delta\epsilon_t)$$
$$\boldsymbol{h}_3^{(t)} = \boldsymbol{w}^* - \boldsymbol{w}^{(t)} + O(\delta\epsilon_t) .$$

Suppose by construction, $\epsilon_t \to 0$ when $t \to \infty$. Then $H_1^{(t)} - h_2^{(t)}I + M^{(t)} \to M^*$ and $\boldsymbol{h}_3^{(t)} + \boldsymbol{w}^{(t)} \to \boldsymbol{w}^*$ and then the proof of Theorem 1 is completed. In the following we only need to show that Lemma 10 constructs an estimation sequence with $\epsilon_t = O(\delta^t) \to 0$. To this end, we need a few things from matrix perturbation theory.

By Theorem 1, $U^{(t)}$ will converge to $U^*$ up to column order perturbation. We use the largest canonical angle to measure the subspace distance spanned by $U^{(t)}$ and $U^*$, which is denoted as $\theta_t = \theta(U^{(t)}, U^*)$. For any matrix $U$, it is well known [Zhu and Knyazev, 2013] that

$$\sin\theta(U, U^*) = \|U_\perp^{*\top}U\|_2, \ \cos\theta(U, U^*) = \sigma_k\{U^{*\top}U\}, \ \tan\theta(U, U^*) = \|U_\perp^{*\top}U(U^{*\top}U)^{-1}\|_2 .$$

The last tangent equality allows us to bound the canonical angle after QR decomposition. Suppose $U^{(t)}R = \hat{U}^{(t)}$ in the QR step of Algorithm 1, we have

$$\tan\theta(\hat{U}^{(t)}, U^*) = \|U_\perp^{*\top}\hat{U}^{(t)}(U^{*\top}\hat{U}^{(t)})^{-1}\|_2 = \|U_\perp^{*\top}U^{(t)}R(U^{*\top}U^{(t)}R)^{-1}\|_2$$
$$= \|U_\perp^{*\top}U^{(t)}(U^{*\top}U^{(t)})^{-1}\|_2 = \tan\theta(U^{(t)}, U^*) .$$

Therefore, it is more convenient to measure the subspace distance by tangent function.

To show $\epsilon_t \to 0$, we recursively define the following variables:

$$\alpha_t \triangleq \tan\theta_t, \ \beta_t \triangleq \|\boldsymbol{w}^* - \boldsymbol{w}^{(t)}\|_2, \ \gamma_t \triangleq \|M^* - M^{(t)}\|_2, \ \epsilon_t \triangleq \beta_t + \gamma_t .$$

The following lemma derives the recursive inequalities regarding to $\{\alpha_t, \beta_t, \gamma_t\}$.

**Lemma 11.** *Under the same settings of Theorem 1, suppose $\alpha_t \leq 2$, $\delta\epsilon_t \leq 4\sqrt{5}\sigma_k^*$, then*

$$\alpha_{t+1} \leq 4\sqrt{5}\delta\sigma_k^{*-1}(\beta_t + \gamma_t), \ \beta_{t+1} \leq \delta(\beta_t + \gamma_t), \ \gamma_{t+1} \leq \alpha_{t+1}\|M^*\|_2 + 2\delta(\beta_t + \gamma_t) .$$

In Lemma 11, when we choose $n$ such that $\delta = O(1/\sqrt{n})$ is small enough, $\{\alpha_t, \beta_t, \gamma_t\}$ will converge to zero. The only question is the initial value $\{\alpha_0, \beta_0, \gamma_0\}$. According to the initialization step of gFM, $\beta_0 \leq \|\boldsymbol{w}^*\|_2$ and $\gamma_0 \leq \|M^*\|_2$. To bound $\alpha_0$, we need the following lemma which directly follows Wely's and Wedin's theorems [Stewart and Sun, 1990].

**Lemma 12.** *Denote $U$ and $\widetilde{U}$ as the top-$k$ left singular vectors of $M$ and $\widetilde{M} = M + O(\epsilon)$ respectively. The $i$-th singular value of $M$ is $\sigma_i$. Suppose that $\epsilon \leq \frac{\sigma_k - \sigma_{k+1}}{4}$. Then the largest canonical angle between $U$ and $\widetilde{U}$, denoted as $\theta(U, \widetilde{U})$, is bounded by $\sin \theta(U, \widetilde{U}) \leq 2\epsilon/(\sigma_k - \sigma_{k+1})$.*

According to Lemma 12, when $2\delta(\|\boldsymbol{w}^*\|_2 + \|M^*\|_2) \leq \sigma_k^*/4$, we have $\sin \theta_0 \leq 4\delta(\|\boldsymbol{w}^*\|_2 + \|M^*\|_2)/\sigma_k^*$. Therefore, $\alpha_0 \leq 2$ provided $\delta \leq \sigma_k^*/[8(\|\boldsymbol{w}^*\|_2 + \|M^*\|_2)]$.

*Proof of Theorem 1.* Suppose that at step $t$, $\alpha_t \leq 2$, $\delta\epsilon_t \leq 4\sqrt{5}\sigma_k^*$, from Lemma 11,

$$\beta_{t+1} + \gamma_{t+1} \leq \beta_{t+1} + \alpha_{t+1}\|M^*\|_2 + 2\delta(\beta_t + \gamma_t) \leq \delta\epsilon_t + 4\sqrt{5}\delta\sigma_k^{*-1}\epsilon_t\|M^*\|_2 + 2\delta\epsilon_t$$
$$= (4\sqrt{5}\sigma_1^*/\sigma_k^* + 3)\delta\epsilon_t .$$

Therefore,

$$\epsilon_t = \beta_t + \gamma_t \leq [(4\sqrt{5}\sigma_1^*/\sigma_k^* + 3)\delta]^t(\beta_0 + \gamma_0)$$
$$\alpha_{t+1} \leq 4\sqrt{5}\delta\sigma_k^{*-1}(\beta_t + \gamma_t) \leq 4\sqrt{5}\delta\sigma_k^{*-1}[(4\sqrt{5}\sigma_1^*/\sigma_k^* + 3)\delta]^t(\beta_0 + \gamma_0) .$$

Clearly we need $(4\sqrt{5}\sigma_1^*/\sigma_k^* + 3)\delta < 1$ to ensure convergence, which is guaranteed by $\delta < \frac{\sigma_k^*}{4\sqrt{5}\sigma_1^* + 3\sigma_k^*}$. To ensure the recursive inequality holds for any $t$, we require $\alpha_{t+1} \leq 2$, which is guaranteed by

$$4\sqrt{5}(\beta_0 + \gamma_0)\delta/\sigma_k^* \leq 2 \Leftrightarrow \delta \leq \frac{\sigma_k^*}{2\sqrt{5}(\sigma_1^* + \beta_0)} .$$

To ensure the condition $\delta\epsilon_t \leq 4\sqrt{5}\sigma_k^*$,

$$\delta \leq 4\sqrt{5}\sigma_k^*/\epsilon_0 = 4\sqrt{5}\sigma_k^*/(\sigma_1^* + \beta_0) \Rightarrow \delta \leq 4\sqrt{5}\sigma_k^*/\epsilon_t .$$

In summary, when

$$\delta \leq \min\left\{\frac{\sigma_k^*}{4\sqrt{5}(\sigma_1^* + \beta_0)}, \frac{\sigma_k^*}{4\sqrt{5}\sigma_1^* + 3\sigma_k^*}, \frac{\sigma_k^*}{2\sqrt{5}(\sigma_1^* + \beta_0)}, \frac{\sigma_k^*}{8(\sigma_1^* + \beta_0)}\right\}$$
$$\Leftarrow \delta \leq \frac{\sigma_k^*}{4\sqrt{5}\sigma_1^* + 3\sigma_k^* + 4\sqrt{5}\beta_0} .$$

we have

$$\epsilon_t = [(4\sqrt{5}\sigma_1^*/\sigma_k^* + 3)\delta]^t(\sigma_1^* + \gamma_0) .$$

To simplify the result, replace $\delta$ with $\delta_1 = (4\sqrt{5}\sigma_1^*/\sigma_k^* + 3)\delta$. The proof is completed. $\square$

## 4.1 Noisy Case

In this subsection, we analyze the performance of gFM under noisy setting. Suppose that $M^*$ is no longer low rank, $M^* = U^*\Lambda^*U^{*\top} + U_\perp^*\Lambda_\perp^*U_\perp^{*\top}$ where $\Lambda_\perp^* = \text{diag}(\lambda_{k+1}, \cdots, \lambda_d)$ is the residual spectrum. Denote $M_k^* = U^*\Lambda^*U^{*\top}$ to be the best rank $k$ approximation of $M^*$ and $M_\perp^* = M^* - M_k^*$. The additive noise $\xi_i$'s are independently sampled from subgaussian with proxy variance $\xi$.

First we generalize the above theorems and lemmas to noisy case.

**Lemma 13.** *Suppose that in Eq. (1) $\boldsymbol{x}_i$'s are independent standard random Gaussian vectors. $M$ is a fixed rank $k$ matrix. $M_\perp^* \neq \boldsymbol{0}$ and $\xi > 0$. Then provided $n \geq Ck^3d/\delta^2$, with a probability at least $1 - \eta$,*

$$\|\frac{1}{2n}\mathcal{A}'\mathcal{A}(M^* - M) - \frac{1}{2}\text{tr}(M_k^* - M)I - (M_k^* - M)\|_2 \leq \delta\|M_k^* - M\|_2 + C\sigma_{k+1}^*d^2/\sqrt{n} \quad (5)$$

$$|\frac{1}{n}\boldsymbol{1}^\top\mathcal{A}(M^* - M) - \text{tr}(M_k^* - M)| \leq \delta\|M_k^* - M\|_2 + C\sigma_{k+1}^*d^2/\sqrt{n} \quad (6)$$

$$\|\frac{1}{n}X^\top\mathcal{A}(M^* - M)\|_2 \leq \delta\|M_k^* - M\|_2 + C\sigma_{k+1}^*d^2/\sqrt{n} \quad (7)$$

$$\|\frac{1}{n}\mathcal{A}'(X^\top\boldsymbol{w})\|_2 \leq \delta\|\boldsymbol{w}\|_2, \ \|\frac{1}{n}\boldsymbol{1}^\top X^\top\boldsymbol{w}\|_2 \leq \delta\|\boldsymbol{w}\|_2 . \quad (8)$$

Define $\gamma_t = \|M_k^* - M^{(t)}\|_2$ similar to the noise-free case. According to Lemma 13, when $\xi = 0$, for $n \geq Ck^3d/\delta^2$,

$$H_1^{(t)} = M_k^* - M^{(t)} + \frac{1}{2}\text{tr}(M_k^* - M^{(t)})I + O(\delta\epsilon_t + C\sigma_{k+1}^*d^2/\sqrt{n})$$

$$h_2^{(t)} = \text{tr}(M^* - M^{(t)}) + O(\delta\epsilon_t + C\sigma_{k+1}^*d^2/\sqrt{n})$$

$$\boldsymbol{h}_3^{(t)} = \boldsymbol{w}^* - \boldsymbol{w}^{(t)} + O(\delta\epsilon_t + C\sigma_{k+1}^*d^2/\sqrt{n}) \, .$$

Define $r = C\sigma_{k+1}^*d^2/\sqrt{n}$. If $\xi > 0$, it is easy to check that the perturbation becomes $\hat{r} = r + O(\xi/\sqrt{n})$. Therefore we uniformly use $r$ to present the perturbation term. The recursive inequalities regarding to the recovery error is constructed in Lemma 14.

**Lemma 14.** *Under the same settings of Lemma 13, define* $\rho \triangleq 2\sigma_{k+1}^*/(\sigma_k^* + \sigma_{k+1}^*)$. *Suppose that at any step* $i$, $0 \leq i \leq t$, $\alpha_i \leq 2$. *When provided* $4\sqrt{5}(\delta\epsilon_t + r) \leq \sigma_k^* - \sigma_{k+1}^*$,

$$\alpha_{t+1} \leq \rho\alpha_t + \frac{4\sqrt{5}}{\sigma_k^* + \sigma_{k+1}^*}\delta\epsilon_t + \frac{4\sqrt{5}}{\sigma_k^* + \sigma_{k+1}^*}r \, , \ \beta_{t+1} \leq \delta\epsilon_t + r \, , \ \gamma_{t+1} \leq \alpha_{t+1}\|M^*\|_2 + 2\delta\epsilon_t + 2r \, .$$

The solution to the recursive inequalities in Lemma 14 is non-trivial. Comparing to the inequalities in Lemma 11, $\alpha_{t+1}$ is bounded by $\alpha_t$ in noisy case. Therefore, if we simply follow Lemma 11 to construct recursive inequality about $\epsilon_t$, we will quickly be overloaded by recursive expansion terms. The key construction of our solution is to bound the term $\alpha_t + 8\sqrt{5}/(\sigma_k^* + \sigma_{k+1}^*)\delta\epsilon_t$. The solution is given in the following theorem.

**Theorem 15.** *Define constants*

$$c = 4\sqrt{5}/(\sigma_k^* + \sigma_{k+1}^*) \, , \ b = 3 + 4\sqrt{5}\sigma_1^*/(\sigma_k^* + \sigma_{k+1}^*) \, , \ q = (1 + \rho)/2 \, .$$

*Then for any* $t \geq 0$,

$$\alpha_t + 2c\delta\epsilon_t \leq q^t\left(2 - \frac{(1+\rho)cr}{1-q}\right) + \frac{(1+\rho)cr}{1-q} \, . \tag{9}$$

*provided*

$$\delta \leq \min\{\frac{1-\rho}{4\rho\sigma_1^*c}, \frac{\rho}{2b}\} \, , \ (2 + c(\sigma_k^* - \sigma_{k+1}^*))\delta\epsilon_0 + r \leq (\sigma_k^* - \sigma_{k+1}^*) \tag{10}$$

$$4\sqrt{5}\left(4 + 2c(\sigma_k^* - \sigma_{k+1}^*)\right)\delta\epsilon_0 + 4\sqrt{5}\left(4 + (\sigma_k^* - \sigma_{k+1}^*)\right)r \leq (\sigma_k^* - \sigma_{k+1}^*)^2 \, .$$

Theorem 15 gives the convergence rate of gFM under noisy settings. We bound $\alpha_t + 2c\delta\epsilon_t$ as the index of recovery error, whose convergence rate is linear. The convergence rate is controlled by $q$, a constant depends on the eigen gap $\sigma_{k+1}^*/\sigma_k^*$. The final recovery error is bounded by $O(r/(1-q))$. Eq. (10) is the small noise condition to ensure the noisy recovery is possible. Generally speaking, learning a $d \times d$ matrix with $O(d)$ samples is an ill-conditioned problem when the target matrix is full rank. The small noise condition given by Eq. (10) essentially says that $M^*$ can be slightly deviated from low rank manifold and the noise shouldn't be too large to blur the spectrum of $M^*$. When the noise is large, Eq. (10) will be satisfied with $n = O(d^2)$ which is the information-theoretical lower bound for recovering a full rank matrix.

## 5 Conclusion

In this paper, we propose a provable efficient algorithm to solve generalized Factorization Machine (gFM) and rank-one matrix sensing. Our method is based on an one-pass alternating updating framework. The proposed algorithm is able to learn gFM within $O(kd)$ memory on steaming data, has linear convergence rate and only requires matrix-vector product implementation. The algorithm takes no more than $O(k^3d\log(1/\epsilon))$ instances to achieve $O(\epsilon)$ recovery error.

**Acknowledgments**

This work was supported in part by research grants from NIH (RF1AG051710) and NSF (III-1421057 and III-1421100).

## Footnotes

[1]Implementation is available from `https://minglin-home.github.io/`

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
