[Supplementary Material · appendix_2017-01-04_14-05-39.pdf]

# A    Preliminary

In this section, we present several important theorems and lemmas in our analysis.

The following concentration inequalities are well known.

**Lemma 16.** *Let $x_i$ be zero-mean sub-Gaussian distribution with variance proxy $\sigma^2$. Denote $S_n = \sum_{i=1}^n a_i x_i$ for a fixed sequence $\{a_i\}$. Then*

$$\Pr(|S_n| > t) \leq 2\exp(-\frac{t^2}{2\sigma^2(\sum_{i=1}^n a_i^2)}) \ .$$

*That is, with a probability at least $1 - \eta$,*

$$|S_n| \leq \sigma\sqrt{\sum_{i=1}^n a_i^2}\sqrt{2\log(2/\eta)} \ .$$

**Corollary 17.** *Let $x_i \sim \mathcal{N}(0,1)$ be a standard Gaussian distribution. Then with a probability at least $1 - \eta$,*

$$\sum_{i=1}^n a_i(x_i^2 - 1) \leq 2\sqrt{\sum_{i=1}^n a_i^2}\sqrt{2\log(2/\eta)} \ .$$

For random matrix, we have matrix concentration inequalities [Tropp, 2015].

**Theorem 18** (Matrix Bernstein's Inequality [Tropp, 2015]). *Suppose $\{S_i\}_{i=1}^n$ are set of independent random matrices of dimension $d_1 \times d_2$,*

$$\|S_i - ES_i\| \leq L \ .$$

*Define*

$$Z = \sum_{i=1}^n S_i, \ \sigma^2 = \frac{1}{n}\max(E\|(Z - EZ)(Z - EZ)^\top\|_2, E\|(Z - EZ)^\top(Z - EZ)\|_2) \ .$$

*The with a probability at least $1 - \delta$, for any $0 < \epsilon < 1$,*

$$\frac{1}{n}\|Z - EZ\|_2 \leq 9\epsilon\sqrt{\log((d_1 + d_2)/\delta)}$$

*provided*

$$n \geq \max(\sigma^2, L)/\epsilon^2 \ .$$

*And for any $n$,*

$$\frac{1}{n}\|Z - EZ\|_2 \leq \frac{4}{3}\frac{L}{n}\log((d_1 + d_2)/\delta) + 3\sqrt{2\frac{\sigma^2}{n}\log((d_1 + d_2)/\delta)} \ .$$

Using matrix Bernstein's inequality, we can bound the covariance estimator.

**Corollary 19** (Matrix Bernstein's Inequality for Covariance Estimator [Tropp, 2015]). *Suppose $\boldsymbol{x}_i \in \mathbb{R}^d, i = 1, 2, \cdots, n$ are independent random variables with zero mean.*

$$\|\boldsymbol{x}_i\|^2 \leq B, \ A = E(\boldsymbol{x}_i\boldsymbol{x}_i^\top)$$

*Then with a probability at least $1 - \delta$,*

$$\|\frac{1}{n}\sum_{i=1}^n \boldsymbol{x}_i\boldsymbol{x}_i^\top - A\|_2 \leq 9\epsilon\sqrt{\log(2d/\delta)/n}$$

*provided*

$$n \geq \max(B\|A\|, B)/\epsilon^2 \ .$$

# B    Proof of Lemmas

## B.1    Proof of Lemma 5

*Proof.* Denote the eigenvalue decomposition of $M$ as

$$M = U\Lambda U^\top = U\text{diag}(\lambda_1, \lambda_2, \cdots, \lambda_k)U^\top$$

Since Gaussian distribution is rotation invariant, $\hat{\boldsymbol{x}}_i = U^\top \boldsymbol{x}_i$ also follows standard Gaussian distribution.

$$\boldsymbol{x}_i{}^\top M \boldsymbol{x}_i = \boldsymbol{x}_i{}^\top U \Lambda U^\top \boldsymbol{x}_i = |\hat{\boldsymbol{x}}_i{}^\top \Lambda \hat{\boldsymbol{x}}_i| = \sum_{j=1}^{k} \lambda_j \hat{\boldsymbol{x}}_{i,j}^2 \ .$$

It is easy to see that $E(\boldsymbol{x}_i{}^\top M \boldsymbol{x}_i) = \sum_j \lambda_j = \mathrm{tr}(M)$. Define

$$a_i \triangleq \boldsymbol{x}_i{}^\top M \boldsymbol{x}_i - \mathrm{tr}(M) = \sum_{j=1}^{d} \lambda_j (\hat{\boldsymbol{x}}_{i,j}^2 - 1)$$

According to Corollary 17, for a fixed $i$, with a probability at least $1 - \eta$,

$$|a_i| \le 2 \|M\|_F \sqrt{2 \log(2/\eta)} \ .$$

Then for any $i$, with a probability at least $1 - \eta$,

$$|a_i| \le 2 \|M\|_F \sqrt{2 \log(2n/\eta)} \ .$$

Apply Corollary 17 again, with a probability at least $1 - 2\eta$,

$$|\frac{1}{n} \sum_{i=1}^{n} a_i - \mathrm{tr}(M)| \le 2 \|M\|_F \sqrt{2 \log(2n/\eta)} \sqrt{2 \log(2/\eta)/n}$$

$$\le 2\sqrt{k} \|M\|_2 \sqrt{2 \log(2n/\eta)} \sqrt{2 \log(2/\eta)/n} \ .$$

Denote $\delta = 2\sqrt{k} \sqrt{2 \log(2n/\eta)} \sqrt{2 \log(2/\eta)/n}$. Then when $n \ge Ck/\delta^2$,

$$|\frac{1}{n} \sum_{i=1}^{n} a_i - \mathrm{tr}(M)| \le \|M\|_2 \delta \ .$$

$\square$

## B.2 Proof of Lemma 6

*Proof.* Define random variable

$$a_i = \boldsymbol{x}_i{}^\top \boldsymbol{w}, \ E a_i = 0$$
$$E a_i^2 \le \|\boldsymbol{w}\|_2^2$$

Then according to Lemma 16, with a probability at least $1 - \eta$,

$$|\frac{1}{n} \sum_{i=1}^{n} a_i| \le \|\boldsymbol{w}\|_2 \sqrt{2 \log(2/\eta)/n} \ .$$

$\square$

## B.3 Proof of Lemma 8

*Proof.* Define random vector

$$\boldsymbol{a}_i = \boldsymbol{x}_i \boldsymbol{x}_i{}^\top M \boldsymbol{x}_i, \ E \boldsymbol{a}_i = 0 \ .$$

With a probability at least $(1 - \eta_1)(1 - \eta_2)$,

$$\|\boldsymbol{a}_i\|_2 = \|\boldsymbol{x}_i \boldsymbol{x}_i{}^\top M \boldsymbol{x}_i\|_2 \le \|\boldsymbol{x}_i{}^\top M \boldsymbol{x}_i\|_2 \|\boldsymbol{x}_i\|_2$$
$$\le (|\mathrm{tr}(M)| + 2\|M\|_F \sqrt{2 \log(2n/\eta_1)}) \sqrt{2d \log(2n/\eta_2)}$$
$$\triangleq c_1 \sqrt{2d \log(2n/\eta_2)} \ .$$

$$\|E \boldsymbol{a}_i{}^\top \boldsymbol{a}_i\|_2 = \|\boldsymbol{x}_i{}^\top M \boldsymbol{x}_i \boldsymbol{x}_i{}^\top \boldsymbol{x}_i \boldsymbol{x}_i{}^\top M \boldsymbol{x}_i\|_2$$
$$\le (\boldsymbol{x}_i{}^\top M \boldsymbol{x}_i)^2 \|\boldsymbol{x}_i\|_2^2$$
$$\le 2 c_1^2 d \log(2n/\eta_2) \ .$$

By matrix Bernstein's inequality, the concentration holds when

$$n \ge \frac{1}{\epsilon^2} \max\{c_1 \sqrt{2d \log(2n/\eta_2)}, 2c_1^2 d \log(2n/\eta_2)\}$$
$$= \frac{1}{\epsilon^2} O(k^2 d \|M\|_2^2) \ .$$

Therefore, after taking the union bound, there exists some constant $C_2 = O(\log(2n/\eta))$,

$$\|\frac{1}{n}\sum_{i=1}^{n}\boldsymbol{a_i}\|_2 \leq \epsilon$$

$$n \geq C_2 k^2 d\|M\|_2^2 \log(2(d+1)/\eta)/\epsilon^2 .$$

Denote $\delta = \|M\|_2/\epsilon$. Then when $n \geq Ck^2d/\delta$,

$$\|\frac{1}{n}\sum_{i=1}^{n}\boldsymbol{a_i}\|_2 \leq \|M\|_2\delta .$$

$\square$

## B.4 Proof of Lemma 7

$$\|\frac{1}{n}\mathcal{A}'(X^\top\boldsymbol{w})\|_2 = \|\frac{1}{n}\sum_{i=1}^{n}\boldsymbol{x}_i\boldsymbol{x}_i^\top\boldsymbol{w}\boldsymbol{x}_i^\top\|_2 .$$

$$E\{\boldsymbol{x}_i\boldsymbol{x}_i^\top\boldsymbol{w}\boldsymbol{x}_i^\top\} = 0$$

$$\begin{aligned}\|\boldsymbol{x}_i\boldsymbol{x}_i^\top\boldsymbol{w}\boldsymbol{x}_i^\top\|_2 &\leq \|\boldsymbol{x}_i^\top\boldsymbol{w}\|_2\|\boldsymbol{x}_i\|_2^2\\ &\leq 2\|\boldsymbol{w}\|_2\sqrt{2\log(2/\eta)}(d+2\sqrt{2d\log(2n/\eta)})\\ &\leq 4\|\boldsymbol{w}\|_2\sqrt{2\log(2/\eta)}d\end{aligned}$$

provided $d \geq 8\log(2n/\eta)$. Now considering

$$\{E\boldsymbol{x}_i\boldsymbol{x}_i^\top\boldsymbol{w}\boldsymbol{x}_i^\top\boldsymbol{x}_i\boldsymbol{w}^\top\boldsymbol{x}_i\boldsymbol{x}_i^\top\}_{pq} = E\{(\sum x_p x_q w_{i1}x_{i1}w_{i2}x_{i2}x_{i3}^2)\}$$

When $p \neq q$,

$$\begin{aligned}E\{(\sum x_p x_q w_{i1}x_{i1}w_{i2}x_{i2}x_{i3}^2)\} &= E\{(2\sum_{i3} x_p x_q w_p x_p w_q x_q x_{i3}^2)\}\\ &= E\{(2\sum_{i3} x_p^2 x_q^2 w_p w_q x_{i3}^2)\}\\ &= 2w_p w_q E\{(\sum_{i3} x_p^2 x_q^2 x_{i3}^2)\}\\ &= 2w_p w_q d\end{aligned}$$

When $p = q$,

$$\begin{aligned}E\{(\sum x_p x_q w_{i1}x_{i1}w_{i2}x_{i2}x_{i3}^2)\} &= E\{(\sum x_p^2 w_{i1}x_{i1}w_{i2}x_{i2}x_{i3}^2)\}\\ &= E\{(\sum x_p^2 w_p x_p w_p x_p x_{i3}^2 + \sum x_p^2 w_j x_j w_j x_j x_{i3}^2 + \sum x_p^2 w_{i3}x_{i3}w_{i3}x_{i3}x_{i3}^2)\}\\ &= E\{(\sum_{i3\neq p} x_p^4 w_p^2 x_{i3}^2 + \sum_{j\neq i3\neq p} x_p^2 w_j^2 x_j^2 x_{i3}^2 + \sum_{i3\neq p} x_p^2 w_{i3}^2 x_{i3}^4)\}\\ &= w_p^2(d-1) + \sum_{j\neq p} w_j^2(d-1) + \sum_{i3\neq p} w_{i3}^2\\ &= w_p^2(d-1) + \sum_{j\neq p} w_j^2 d = w_p^2(d-1) + \sum_{j=1}^{d} w_j^2 d - w_p^2 d\\ &= \sum_{j=1}^{d} w_j^2 d - w_p^2\end{aligned}$$

Therefore,

$$E\boldsymbol{x}_i\boldsymbol{x}_i^\top\boldsymbol{w}\boldsymbol{x}_i^\top\boldsymbol{x}_i\boldsymbol{w}^\top\boldsymbol{x}_i\boldsymbol{x}_i^\top = d\,\mathrm{diag}\{\|\boldsymbol{w}\|_2^2\} - \mathrm{diag}\{\boldsymbol{w}\circ\boldsymbol{w}\} + 2d\boldsymbol{w}\boldsymbol{w}^\top$$

$$\|E\boldsymbol{x}_i\boldsymbol{x}_i^\top\boldsymbol{w}\boldsymbol{x}_i^\top\boldsymbol{x}_i\boldsymbol{w}^\top\boldsymbol{x}_i\boldsymbol{x}_i^\top\|_2 \leq 4d\|\boldsymbol{w}\|_2^2$$

Using matrix Bernstein's inequality,

$$\|\frac{1}{n}\sum_{i=1}^{n} \boldsymbol{x}_i\boldsymbol{x}_i^\top \boldsymbol{w}\boldsymbol{x}_i^\top\|_2 \leq \frac{4}{3}\frac{4\|\boldsymbol{w}\|_2\sqrt{2\log(2/\eta)}d}{n}\log(2d/\eta)$$

$$+ 3\sqrt{2\frac{4d\|\boldsymbol{w}\|_2^2}{n}\log(2d/\eta)}$$

$$\leq C\|\boldsymbol{w}\|_2\sqrt{\frac{d}{n}}$$

Denote $\delta = C\sqrt{d/n}$, when $n \geq Cd/\delta^2$, $d \geq 8\log(2n/\eta)$,

$$\|\frac{1}{n}\sum_{i=1}^{n} \boldsymbol{x}_i\boldsymbol{x}_i^\top \boldsymbol{w}\boldsymbol{x}_i^\top\|_2 \leq \|\boldsymbol{w}\|_2\delta$$

## B.5  Proof of Lemma 9

According to Corollary 19, when $d \geq 8\log(2n/\eta)$,

$$\|\boldsymbol{x}_i\|^2 \leq 2d$$

Therefore, with a probability at least $1 - \eta$,

$$\|I - \frac{1}{n}XX^\top\|_2 \leq 9\epsilon\sqrt{\log(2d/\eta)/n}$$

for $n \geq 2d/\epsilon^2$. Denote $\delta = 9\epsilon\sqrt{\log(2d/\eta)/n}$, then when $n \geq Cd/\delta^2$,

$$\|I - \frac{1}{n}XX^\top\|_2 \leq \delta .$$

## B.6  Proof of Lemma 11

To derive $\alpha_{t+1}$ ,

$$\|U_\perp^{*\top}[M^* + O(2\delta\epsilon_t)]U^{(t)}\|_2 \leq \|U_\perp^{*\top}M^*U^{(t)}\|_2 + 2\delta\epsilon_t$$
$$\leq 2\delta\epsilon_t$$

$$\sigma_k\{U^{*\top}[M^* + O(2\delta\epsilon_t)]U^{(t)}\} \geq U^{*\top}M^*U^{(t)} - 2\delta\epsilon_t$$
$$\geq \sigma_k^*\sigma_k\{U^{*\top}U^{(t)}\} - 2\delta\epsilon_t$$
$$= \sigma_k^*\cos\theta_t - 2\delta\epsilon_t$$

$$\alpha_{t+1} = \tan\theta_{t+1} = \frac{\|U_\perp^{*\top}[M^* + O(2\delta\epsilon_t)]U^{(t)}\|_2}{\sigma_k\{U^{*\top}[M^* + O(2\delta\epsilon_t)]U^{(t)}\}}$$
$$\leq \frac{2\delta\epsilon_t}{\sigma_k^*\cos\theta_t - 2\delta\epsilon_t} .$$

According to the assumption, $\cos\theta_t \geq \frac{1}{\sqrt{5}}$, $2\delta\epsilon_t \leq \frac{1}{2\sqrt{5}}\sigma_k^*$, therefore

$$\alpha_{t+1} \leq 2\sqrt{5}\epsilon_t/\sigma_k^* = 4\sqrt{5}\delta(\beta_t + \gamma_t)/\sigma_k^* .$$

To derive $\gamma_{t+1}$,

$$\gamma_{t+1} = \|M^* - M^{(t+1)}\|_2$$
$$= \|M^* - (U^{(t+1)}U^{(t+1)\top}(H_1^{(t)} - D(\boldsymbol{h_2}^{(t)}) + M^{(t)})^\top)\|_2$$
$$= \|M^* - U^{(t+1)}U^{(t+1)\top}(M^* + O(2\delta(\gamma_t + \beta_t)))^\top\|_2$$
$$= \|(I - U^{(t+1)}U^{(t+1)\top})M^* + U^{(t+1)}U^{(t+1)\top}O(2\delta(\gamma_t + \beta_t)))^\top\|_2$$
$$\leq \|(I - U^{(t+1)}U^{(t+1)\top})M^*\|_2 + O(2\delta(\gamma_t + \beta_t))$$
$$\leq \tan\theta_{t+1}\|M^*\|_2 + 2\delta(\gamma_t + \beta_t)$$
$$= \alpha_{t+1}\|M^*\|_2 + 2\delta(\gamma_t + \beta_t) .$$

The recursive inequality of $\beta_t$ is trivial.

# C Proof of Theorem 4

*Proof.* Denote $\sigma_1 = \|M\|_2$. Define random matrix

$$B_i = \boldsymbol{x}_i\boldsymbol{x}_i^\top M \boldsymbol{x}_i\boldsymbol{x}_i^\top \ .$$

It is easy to check that

$$EB_i = 2M + \operatorname{tr}(M)I \ .$$

$$\begin{aligned}
\|B_i - EB_i\|_2 &= \|\boldsymbol{x}_i\boldsymbol{x}_i^\top M \boldsymbol{x}_i\boldsymbol{x}_i^\top - 2M - \operatorname{tr}(M)I\|_2 \\
&\leq \|\boldsymbol{x}_i\boldsymbol{x}_i^\top M \boldsymbol{x}_i\boldsymbol{x}_i^\top\|_2 + \|2M - \operatorname{tr}(M)I\|_2 \\
&\leq \|\boldsymbol{x}_i\boldsymbol{x}_i^\top M \boldsymbol{x}_i\boldsymbol{x}_i^\top\|_2 + 2\|M\|_2 + |\operatorname{tr}(M)| \ .
\end{aligned}$$

According to Lemma 5, with a probability at least $1 - \eta_2$, for any $i \in \{1, \cdots, n\}$,

$$|\boldsymbol{x}_i^\top M \boldsymbol{x}_i| \leq |\operatorname{tr}(M)| + 2\|M\|_F \sqrt{2\log(2n/\eta_2)} \triangleq c_1 \ .$$

Therefore we have, with a probability at least $(1 - \eta_1)(1 - \eta_2)$,

$$\begin{aligned}
\|B_i - EB_i\|_2 &\leq \|\boldsymbol{x}_i\|_2^2 \, |\boldsymbol{x}_i^\top M \boldsymbol{x}_i| + 2\|M\|_2 + |\operatorname{tr}(M)| \\
&\leq 2d\log(2n/\eta_1)|\operatorname{tr}(M)| + 2\|M\|_F\sqrt{2\log(2n/\eta_2)} + 2\|M\|_2 + |\operatorname{tr}(M)| \\
&\leq Cdk\sigma_1 \ .
\end{aligned}$$

Next we need to bound

$$\begin{aligned}
\|E(B_i - EB_i)(B_i - EB_i)^\top\|_2 = \|E(B_i^2) - (EB_i)^2\|_2 &\leq \|E(B_i^2)\|_2 + \|EB_i\|_2^2 \\
&\leq \|E(B_i^2)\|_2 + 2|\operatorname{tr}(M)|^2 + 2\|M\|_2^2
\end{aligned}$$

To bound $\|E(B_i^2)\|_2$, denote the eigenvalue decomposition of $M$ as

$$M = U\Lambda U^\top = U\operatorname{diag}(\lambda_1, \lambda_2, \cdots, \lambda_k)U^\top$$

Let $U_\perp$ be the complementary basis matrix of $U$. Define random variables $\boldsymbol{u}_i \triangleq U^\top\boldsymbol{x}_i$, $\boldsymbol{v}_i \triangleq U_\perp^\top\boldsymbol{x}_i$. Since $\boldsymbol{x}_i$ are standard random Gaussian, $\boldsymbol{u}$ and $\boldsymbol{v}$ should also be jointly random Gaussian thus independent.

$$\begin{aligned}
\|E(B_i^2)\|_2 =& \|E(\boldsymbol{x}_i\boldsymbol{x}_i^\top M \boldsymbol{x}_i\boldsymbol{x}_i^\top \boldsymbol{x}_i\boldsymbol{x}_i^\top M \boldsymbol{x}_i\boldsymbol{x}_i^\top)\|_2 \\
=& \left\| E\left( \begin{bmatrix} \boldsymbol{u}_i \\ \boldsymbol{v}_i \end{bmatrix} \boldsymbol{u}_i^\top \Lambda \boldsymbol{u}_i (\|\boldsymbol{u}_i\|_2^2 + \|\boldsymbol{v}_i\|_2^2) \boldsymbol{u}_i^\top \Lambda \boldsymbol{u}_i \begin{bmatrix} \boldsymbol{u}_i \\ \boldsymbol{v}_i \end{bmatrix}^\top \right) \right\|_2 \\
\leq& \|E(\boldsymbol{u}_i\boldsymbol{u}_i^\top \Lambda \boldsymbol{u}_i(\|\boldsymbol{u}_i\|_2^2 + \|\boldsymbol{v}_i\|_2^2)\boldsymbol{u}_i^\top \Lambda \boldsymbol{u}_i\boldsymbol{u}_i^\top)\|_2 \\
&+ 2\|E(\boldsymbol{u}_i\boldsymbol{u}_i^\top \Lambda \boldsymbol{u}_i(\|\boldsymbol{u}_i\|_2^2 + \|\boldsymbol{v}_i\|_2^2)\boldsymbol{u}_i^\top \Lambda \boldsymbol{u}_i\boldsymbol{v}_i^\top)\|_2 \\
&+ \|E(\boldsymbol{v}_i\boldsymbol{u}_i^\top \Lambda \boldsymbol{u}_i(\|\boldsymbol{u}_i\|_2^2 + \|\boldsymbol{v}_i\|_2^2)\boldsymbol{u}_i^\top \Lambda \boldsymbol{u}_i\boldsymbol{v}_i^\top)\|_2 \\
\leq& \|E(\boldsymbol{u}_i\boldsymbol{u}_i^\top \Lambda \boldsymbol{u}_i\|\boldsymbol{u}_i\|_2^2\boldsymbol{u}_i^\top \Lambda \boldsymbol{u}_i\boldsymbol{u}_i^\top)\|_2 + \|E(\boldsymbol{u}_i\boldsymbol{u}_i^\top \Lambda \boldsymbol{u}_i\|\boldsymbol{v}_i\|_2^2\boldsymbol{u}_i^\top \Lambda \boldsymbol{u}_i\boldsymbol{u}_i^\top)\|_2 \\
&+ 2\|E(\boldsymbol{u}_i\boldsymbol{u}_i^\top \Lambda \boldsymbol{u}_i\|\boldsymbol{u}_i\|_2^2\boldsymbol{u}_i^\top \Lambda \boldsymbol{u}_i\boldsymbol{v}_i^\top)\|_2 + 2\|E(\boldsymbol{u}_i\boldsymbol{u}_i^\top \Lambda \boldsymbol{u}_i\|\boldsymbol{v}_i\|_2^2\boldsymbol{u}_i^\top \Lambda \boldsymbol{u}_i\boldsymbol{v}_i^\top)\|_2 \\
&+ \|E(\boldsymbol{v}_i\boldsymbol{u}_i^\top \Lambda \boldsymbol{u}_i\|\boldsymbol{u}_i\|_2^2\boldsymbol{u}_i^\top \Lambda \boldsymbol{u}_i\boldsymbol{v}_i^\top)\|_2 + \|E(\boldsymbol{v}_i\boldsymbol{u}_i^\top \Lambda \boldsymbol{u}_i\|\boldsymbol{v}_i\|_2^2\boldsymbol{u}_i^\top \Lambda \boldsymbol{u}_i\boldsymbol{v}_i^\top)\|_2 \ .
\end{aligned}$$

Let us bound the above 6 terms respectively. Recall that with a probability at least $1 - \eta_2$,

$$|\boldsymbol{u}_i^\top \Lambda \boldsymbol{u}_i| = |\boldsymbol{x}_i^\top M \boldsymbol{x}_i| \leq c_1 \ .$$

With a probability at least $1 - \eta_3$, for any $i \in \{1, \cdots, n\}$, $\|\boldsymbol{u}_i\|_2 \leq 2\sqrt{k\log(4n/\eta_3)}, \|\boldsymbol{v}_i\|_2 \leq 2\sqrt{d\log(4n/\eta_3)}$. Then:

$$\begin{aligned}
& \|E(\boldsymbol{u}_i\boldsymbol{u}_i^\top \Lambda \boldsymbol{u}_i\|\boldsymbol{u}_i\|_2^2\boldsymbol{u}_i^\top \Lambda \boldsymbol{u}_i\boldsymbol{u}_i^\top)\|_2 \\
=& \|E\{\left((\boldsymbol{u}_i^\top \Lambda \boldsymbol{u}_i)^2\|\boldsymbol{u}_i\|_2^2\right)\boldsymbol{u}_i\boldsymbol{u}_i^\top\}\|_2 \\
\leq& (\boldsymbol{u}_i^\top \Lambda \boldsymbol{u}_i)^2\|\boldsymbol{u}_i\|_2^4 \\
\leq& 32c_1^2k^2\log^2(2n/\eta_3) \ .
\end{aligned}$$

$$\|E(\boldsymbol{u}_i\boldsymbol{u}_i^\top\Lambda\boldsymbol{u}_i\|\boldsymbol{v}_i\|_2^2\boldsymbol{u}_i^\top\Lambda\boldsymbol{u}_i\boldsymbol{u}_i^\top)\|_2$$
$$=\|E(\|\boldsymbol{v}_i\|_2^2)E(\boldsymbol{u}_i\boldsymbol{u}_i^\top\Lambda\boldsymbol{u}_i\boldsymbol{u}_i^\top\Lambda\boldsymbol{u}_i\boldsymbol{u}_i^\top)\|_2$$
$$\leq 4d\log(4n/\eta_3)\|E(\boldsymbol{u}_i\boldsymbol{u}_i^\top\Lambda\boldsymbol{u}_i\boldsymbol{u}_i^\top\Lambda\boldsymbol{u}_i\boldsymbol{u}_i^\top)\|_2$$
$$\leq 4d\log(4n/\eta_3)\|\boldsymbol{u}_i\|_2^2(\boldsymbol{u}_i^\top\Lambda\boldsymbol{u}_i)^2$$
$$\leq 4d\log(4n/\eta_3)c_1^2(4k\log(4n/\eta_3))\ .$$

$$2\|E(\boldsymbol{u}_i\boldsymbol{u}_i^\top\Lambda\boldsymbol{u}_i\|\boldsymbol{u}_i\|_2^2\boldsymbol{u}_i^\top\Lambda\boldsymbol{u}_i\boldsymbol{v}_i^\top)\|_2$$
$$=2\|E(\boldsymbol{u}_i\boldsymbol{u}_i^\top\Lambda\boldsymbol{u}_i\|\boldsymbol{u}_i\|_2^2\boldsymbol{u}_i^\top\Lambda\boldsymbol{u}_i)E(\boldsymbol{v}_i^\top)\|_2 = 0$$

$$2\|E(\boldsymbol{u}_i\boldsymbol{u}_i^\top\Lambda\boldsymbol{u}_i\|\boldsymbol{v}_i\|_2^2\boldsymbol{u}_i^\top\Lambda\boldsymbol{u}_i\boldsymbol{v}_i^\top)\|_2$$
$$=2\|E(\boldsymbol{u}_i(\boldsymbol{u}_i^\top\Lambda\boldsymbol{u}_i)^2)E(\|\boldsymbol{v}_i\|_2^2\boldsymbol{v}_i^\top)\|_2$$
$$=2\|E(\boldsymbol{u}_i(\boldsymbol{u}_i^\top\Lambda\boldsymbol{u}_i)^2)E(\boldsymbol{v}_i^\top\boldsymbol{v}_i\boldsymbol{v}_i^\top)\|_2 = 0$$

$$\|E(\boldsymbol{v}_i\boldsymbol{u}_i^\top\Lambda\boldsymbol{u}_i\|\boldsymbol{u}_i\|_2^2\boldsymbol{u}_i^\top\Lambda\boldsymbol{u}_i\boldsymbol{v}_i^\top)\|_2$$
$$=\|E(\boldsymbol{u}_i^\top\Lambda\boldsymbol{u}_i\|\boldsymbol{u}_i\|_2^2\boldsymbol{u}_i^\top\Lambda\boldsymbol{u}_i)E(\boldsymbol{v}_i\boldsymbol{v}_i^\top)\|_2$$
$$=\|E(\boldsymbol{u}_i^\top\Lambda\boldsymbol{u}_i\|\boldsymbol{u}_i\|_2^2\boldsymbol{u}_i^\top\Lambda\boldsymbol{u}_i)\|_2$$
$$\leq(\boldsymbol{u}_i^\top\Lambda\boldsymbol{u}_i)^2\|\boldsymbol{u}_i\|_2^2$$
$$\leq 4c_1^2k\log(4n/\eta_3)\ .$$

$$\|E(\boldsymbol{v}_i\boldsymbol{u}_i^\top\Lambda\boldsymbol{u}_i\|\boldsymbol{v}_i\|_2^2\boldsymbol{u}_i^\top\Lambda\boldsymbol{u}_i\boldsymbol{v}_i^\top)\|_2$$
$$=\|E\{(\boldsymbol{u}_i^\top\Lambda\boldsymbol{u}_i)^2\}E(\boldsymbol{v}_i\|\boldsymbol{v}_i\|_2^2\boldsymbol{v}_i^\top)\|_2$$
$$=\|E\{(\boldsymbol{u}_i^\top\Lambda\boldsymbol{u}_i)^2\}(d+2)I\|_2$$
$$\leq(d+2)(\boldsymbol{u}_i^\top\Lambda\boldsymbol{u}_i)^2$$
$$\leq c_1^2(d+2)$$

Add all above together, we have

$$\|E(B_i^2)\|_2 \leq 32c_1^2k^2\log^2(2n/\eta_3) + 4d\log(4n/\eta_3)c_1^2(4k\log(4n/\eta_3))$$
$$+ 4c_1^2k\log(4n/\eta_3) + c_1^2(d+2)$$
$$\leq Ck^3d\sigma_1\ .$$

Apply matrix Bernsterin's inequality, the proof is completed. $\qquad\square$

# D Proof of Lemma 13

We assume that $n \geq Ck^3d/\delta^2$ .

To prove Eq. (5)

$$\|\frac{1}{2n}\mathcal{A}'\mathcal{A}(M^* - M) - \frac{1}{2}\mathrm{tr}(M_k^* - M)I - (M_k^* - M)\|_2$$
$$\leq\|\frac{1}{2n}\mathcal{A}'\mathcal{A}(M_k^* - M) - \frac{1}{2}\mathrm{tr}(M_k^* - M)I - (M_k^* - M)\|_2 + \|\frac{1}{2n}\mathcal{A}'\mathcal{A}(M_\perp^*)\|_2$$
$$\leq\|\frac{1}{2n}\mathcal{A}'\mathcal{A}(M_\perp^*)\|_2 + \delta\|M_k^* - M\|_2\ .$$

The last inequality is because of Theorem 4. To bound the first term in the last inequality, define random matrix

$$B_i = \boldsymbol{x}_i\boldsymbol{x}_i^\top M_\perp^*\boldsymbol{x}_i\boldsymbol{x}_i^\top$$

As proved in Theorem 4, $EB_i = 2M_\perp^* + \mathrm{tr}(M_\perp^*)I$.

$$\|(B_i - EB_i)\|_2 =\|\boldsymbol{x}_i\boldsymbol{x}_i^\top M_\perp^*\boldsymbol{x}_i\boldsymbol{x}_i^\top - 2M_\perp^* + \mathrm{tr}(M_\perp^*)I\|_2$$
$$\leq\|\boldsymbol{x}_i\boldsymbol{x}_i^\top M_\perp^*\boldsymbol{x}_i\boldsymbol{x}_i^\top\|_2 + 2\|M_\perp^*\|_2 + \|\mathrm{tr}(M_\perp^*)I\|_2$$
$$=\|\boldsymbol{x}_i\boldsymbol{x}_i^\top M_\perp^*\boldsymbol{x}_i\boldsymbol{x}_i^\top\|_2 + 2\sigma_{k+1}^* + |\mathrm{tr}(M_\perp^*)|$$

While

$$\|\boldsymbol{x}_i\boldsymbol{x}_i^\top M_\perp^* \boldsymbol{x}_i\boldsymbol{x}_i^\top\|_2 \leq \|M_\perp^*\|_2\|\boldsymbol{x}_i\|_2^4$$
$$\leq \sigma_{k+1}^*(d + 2\sqrt{2d\log(2n/\eta)})^2$$
$$\leq Cd^2\sigma_{k+1}^*$$

Applying matrix Bernstein's inequality, with a probability at least $1 - \eta$, we have

$$\|\frac{1}{n}\sum_{i=1}^{n}(B_i - EB_i)\|_2 \leq C\sigma_{k+1}^*d^2/\sqrt{n} \ .$$

Therefore

$$\|\frac{1}{2n}\mathcal{A}'\mathcal{A}(M^* - M) - \frac{1}{2}\mathrm{tr}(M_k^* - M)I - (M_k^* - M)\|_2 \leq \delta\|M_k^* - M\|_2 + C\sigma_{k+1}^{*2}d^4/\sqrt{n} \ .$$

The other inequalities can be similarly proved.

## E   Proof of Lemma 14

First we bound $\alpha_{t+1}$. According to assumption, when

$$2(\delta\epsilon_t + r) \leq \frac{\sigma_k^* - \sigma_{k+1}^*}{2\sigma_k^*}\sigma_k^*/\sqrt{5}$$

we have

$$\alpha_{t+1} \leq \frac{\sigma_{k+1}^* \sin\theta_t + 2(\delta\epsilon_t + r)}{\sigma_k^* \cos\theta_t - 2(\delta\epsilon_t + r)}$$
$$\leq \frac{2\sigma_k^*}{\sigma_k^* + \sigma_{k+1}^*}\frac{\sigma_{k+1}^* \sin\theta_t + 2(\delta\epsilon_t + r)}{\sigma_k^* \cos\theta_t}$$
$$\leq \frac{2\sigma_{k+1}^*}{\sigma_k^* + \sigma_{k+1}^*}\tan\theta_t + \frac{2}{\sigma_k^* + \sigma_{k+1}^*}\frac{2(\delta\epsilon_t + r)}{\cos\theta_t}$$
$$\leq \frac{2\sigma_{k+1}^*}{\sigma_k^* + \sigma_{k+1}^*}\tan\theta_t + \frac{4\sqrt{5}}{\sigma_k^* + \sigma_{k+1}^*}(\delta\epsilon_t + r)$$
$$\leq \rho\alpha_t + \frac{4\sqrt{5}}{\sigma_k^* + \sigma_{k+1}^*}\delta\epsilon_t + \frac{4\sqrt{5}}{\sigma_k^* + \sigma_{k+1}^*}r \ .$$

To bound $\beta_{t+1}$. Clearly $\beta_{t+1} \leq \delta\epsilon_t + r$.

To bound $\gamma_{t+1}$, following the noise-free case,

$$\gamma_{t+1} \leq \alpha_{t+1}\|M^*\|_2 + 2\delta\epsilon_t + 2r \ .$$

## F   Proof of Lemma 15

Abbreviate

$$c = \frac{4\sqrt{5}}{\sigma_k^* + \sigma_{k+1}^*}$$

Then

$$\alpha_{t+1} \leq \rho\alpha_t + c\delta\epsilon_t + cr \ .$$

According to Lemma 14,

$$\beta_{t+1} + \gamma_{t+1} \leq \delta\epsilon_t + r + \alpha_{t+1}\|M^*\|_2 + 2\delta\epsilon_t + 2r$$
$$= \sigma_1^*\alpha_{t+1} + 3\delta\epsilon_t + 3r$$
$$\leq \sigma_1^*(\rho\alpha_t + c\delta\epsilon_t + cr) + 3\delta\epsilon_t + 3r$$
$$= \rho\sigma_1^*\alpha_t + (\sigma_1^*c + 3)\delta\epsilon_t + (\sigma_1^*c + 3)r$$

Therefore, abbreviate $b \triangleq (\sigma_1^*c + 3)$,

$$\begin{cases} \alpha_{t+1} \leq \rho\alpha_t + c\delta\epsilon_t + cr \\ \epsilon_{t+1} \leq \rho\sigma_1^*\alpha_t + b\delta\epsilon_t + br \end{cases}$$

define
$$f_t = \alpha_t + 2c\delta\epsilon_t$$

$$
\begin{aligned}
f_{t+1} &= a_{t+1} + 2c\delta\epsilon_{t+1} \\
&\leq \rho\alpha_t + c\delta\epsilon_t + cr + 2c\delta(\rho\sigma_1^*\alpha_t + b\delta\epsilon_t + br) \\
&= \rho\alpha_t + c\delta\epsilon_t + cr + 2c\delta\rho\sigma_1^*\alpha_t + 2c\delta b\delta\epsilon_t + 2c\delta br \\
&= (\rho + 2c\delta\rho\sigma_1^*)\alpha_t + (c + 2c\delta b)\delta\epsilon_t + (1 + 2\delta b)cr
\end{aligned}
$$

When
$$\delta \leq \frac{1-\rho}{4\rho\sigma_1^* c}$$
$$\Rightarrow \rho + 2c\delta\rho\sigma_1^* \leq \frac{1+\rho}{2}$$

And when
$$\Rightarrow \delta \leq \frac{\rho}{2b}$$
$$\Rightarrow 2\delta b \leq \rho$$
$$\Rightarrow 2c\delta b \leq \rho c$$
$$\Rightarrow c + 2c\delta b \leq (1+\rho)c$$
$$\Rightarrow c + 2c\delta b \leq \frac{1+\rho}{2}2c$$

Then abbreviate $R \triangleq (c + 2c\delta b)\delta\epsilon_t + (1 + 2\delta b)cr$ we have
$$f_{t+1} \leq \frac{1+\rho}{2}f_t + (1 + 2\delta b)cr \leq \frac{1+\rho}{2}f_t + (1+\rho)cr$$

Abbreviate $q = (1+\rho)/2$,
$$f_t \leq \frac{(1+\rho)cr}{1-q} + q^t\left(f_0 - \frac{(1+\rho)cr}{1-q}\right)$$

To ensure $\alpha_{t+1} \leq 2$, we require
$$f_0 \leq 2$$
$$\Leftarrow \alpha_0 + 2c\delta\epsilon_0 \leq 2$$

According to Lemma 12,
$$\alpha_0 \leq \frac{2}{\sigma_k^* - \sigma_{k+1}^*}2(\delta\epsilon_0 + r) = \frac{4}{\sigma_k^* - \sigma_{k+1}^*}(\delta\epsilon_0 + r)$$

$$
\begin{aligned}
&\alpha_0 + 2c\delta\epsilon_0 \leq 2 \\
\Leftarrow & \frac{4}{\sigma_k^* - \sigma_{k+1}^*}(\delta\epsilon_0 + r) + 2c\delta\epsilon_0 \leq 2 \\
\Leftarrow & (4 + 2c(\sigma_k^* - \sigma_{k+1}^*))\delta\epsilon_0 + 4r \leq 2(\sigma_k^* - \sigma_{k+1}^*) \\
\Leftarrow & (2 + c(\sigma_k^* - \sigma_{k+1}^*))\delta\epsilon_0 + r \leq (\sigma_k^* - \sigma_{k+1}^*)
\end{aligned}
$$

In summary,
$$\alpha_t + 2c\delta\epsilon_t \leq q^t\left(f_0 - \frac{(1+\rho)cr}{1-q}\right) + \frac{(1+\rho)cr}{1-q}$$

provided
$$\delta \leq \min\left\{\frac{1-\rho}{4\rho\sigma_1^* c}, \frac{\rho}{2b}\right\}$$

and
$$(2 + c(\sigma_k^* - \sigma_{k+1}^*))\delta\epsilon_0 + r \leq (\sigma_k^* - \sigma_{k+1}^*)$$
$$4\sqrt{5}(\delta\max_t \epsilon_t + r) \leq \sigma_k^* - \sigma_{k+1}^*$$

To ensure the last inequality,

$$\delta \max_t \epsilon_t \le f_0 \le \alpha_0 + 2c\delta\epsilon_0 \le \frac{4}{\sigma_k^* - \sigma_{k+1}^*}(\delta\epsilon_0 + r) + 2c\delta\epsilon_0$$

$$= (\frac{4}{\sigma_k^* - \sigma_{k+1}^*} + 2c)\delta\epsilon_0 + \frac{4}{\sigma_k^* - \sigma_{k+1}^*}r$$

Therefore we need the condition

$$4\sqrt{5}\left(\frac{4}{\sigma_k^* - \sigma_{k+1}^*} + 2c\right)\delta\epsilon_0 + 4\sqrt{5}\left(\frac{4}{\sigma_k^* - \sigma_{k+1}^*} + 1\right)r \le \sigma_k^* - \sigma_{k+1}^*$$

$$\Leftarrow 4\sqrt{5}\left(4 + 2c(\sigma_k^* - \sigma_{k+1}^*)\right)\delta\epsilon_0 + 4\sqrt{5}\left(4 + (\sigma_k^* - \sigma_{k+1}^*)\right)r \le (\sigma_k^* - \sigma_{k+1}^*)^2$$