[Reviews · NeurIPS 2016]

Reviewer 1

Summary

This paper considers the following problem: given measurements y_i =x_i^T w + x_i^T M x_i, recover w, M assuming M is low rank. Without the linear term, this is just rank one matrix sensing and there have been some recent work studying that problem. This paper considers a moment based alternating minimization method to solve this problem. High level idea is to estimate w from X (y -x \sum_i x_i^T M x_i ) + XX' w_t, so if current estimate of M, M_t is close to M, then the above estimate will be close to w. standard spectral initialization is used to set initial M_0. Convergence results are provided for the alternating minimization method, assuming independent samples are present in each iteration with a sample complexity of O(k^3 d ) per iteration.

Qualitative Assessment

1) The conditionally independent RIP is just the standard RIP assumption with resampling in each iterate of the algorithm. This is a standard trick used in many recent papers to avoid dependency between iterates in the analysis and there is no need to call it a new definition, as it is hard to imagine this assumption will hold under any other setting than simple resampling. 2) The paper currently lacks any simulations results. It would have been nice to see the performance of the algorithm on atleast some synthetic data. 3) Regarding the sample complexity of O(k^3 d), there are some works on analyzing gradient descent for matrix sensing problem to get the optimal sample complexity of O(kd)(http://arxiv.org/abs/1507.03566, http://arxiv.org/abs/1509.03917). I think it can be extended to this setting too. 4) Typos in lines 23, 124, Algorithm (line 1, what is y*?)

Confidence in this Review

2-Confident (read it all; understood it all reasonably well)


Reviewer 2

Summary

The authors approach the analysis of quadratic factorization machines from the standpoint of a matrix estimation using measurements consisting of a linear + quadratic term. This differs from the usual case in the literature where the measurements are only quadratic (a trace product with a rank-one sensing matrix). The authors introduce an algorithm for this case, and present theory that it converges to the true model when the measurements are Gaussian.

Qualitative Assessment

This model is not actually a factorization machine, as it allows for self interaction terms (i.e., M is not hollow as it is in the case of FMs). One of the key features of FMs is the lack of self-interaction terms, and real data doesn't meet the Gaussian assumptions (or RIP assumptions) used in the theory, so experiments would have been very helpful to gauge the usefulness of this model. That said, the analysis is exciting, because it shows how to handle a model that has both linear and quadratic terms, and this paper represents a solid contribution to the literature on the analysis of the convergence of non-convex algorithms.

Confidence in this Review

2-Confident (read it all; understood it all reasonably well)


Reviewer 3

Summary

This paper proposes a new algorithm for training a variant of Factorization Machines (FMs) (second-order polynomial parameterized by a low-rank matrix). Pros: - simple, efficient algorithm - novel convergence/recovery analysis *assuming* Gaussian-distributed data Con: - no experimental validation at all Note that the model considered in this paper is actually not the original FMs but a variant (some authors call it "polynomial networks"; see below). The authors might want to update the title and main body of their paper to reflect this.

Qualitative Assessment

Major comments -------------- * An obvious major issue with this paper is the lack of experiments. How does the algorithm compare to gradient-based local search algorithms? I would be curious to see if it works better on i) Gaussian distributed data and ii) real data. Even if the results turn out to be similar, perhaps the authors can find some advantages to their algorithm such as, e.g., robustness to initialization. * Technically, the model considered in this paper is a *variant* of FMs in two ways 1) The model considered uses the quadratic form x^T M x = \sum_{i=1}^d \sum_{j=1}^d M_{i,j} x_i x_j while FMs use \sum_{i=1}^{d-1} \sum_{j=i+1}^d M_{i,j} x_i x_j. Recent works [1, 2] have called this variant "polynomial network" so the authors might want to use this name instead of "factorization machine". If the proposed algorithm and analysis trivially extends to the original FMs, the authors might want to add a few words about this as well. [1] https://arxiv.org/abs/1410.1141 [2] http://jmlr.org/proceedings/papers/v48/blondel16.html 2) The original FMs factor M as V V^T, hence M is p.s.d. It would be interesting to briefly discuss this point. In particular, it is not clear whether a p.s.d. constraint is needed or whether it can hurt performance. * Theorem 1 assumes that the data is Gaussian distributed. This should be stated clearly in the abstract and introduction too. Detailed comments ----------------- * Lines 80-81: > When d is much larger than k, convex programming on the trace norm or nuclear norm of M∗ becomes intractable since M ∗ is of d × d. This claim is wrong. Modern solvers for nuclear norm penalized minimization have no problem scaling to large d provided that the data is sparse. The authors should make a weaker claim or a more precise statement. For example, "although the estimation of M* can be formulated as convex programming based on the nuclear norm, a more popular strategy in practice is to ...". * Lines 113-116: You could also use \hat{x}_i^T M \hat{x}_i (i.e., use \hat{x}_i both on the left and right). * How difficult is it to extend the proposed algorithm to other losses than the squared loss? * Line 123: Due to the name "one-pass", I assume the instances are sampled without replacement? * Algorithm 1: H_3 seems to be a vector so it would be less confusing to not capitalize it. * The precise objective function minimized in this paper does not seem to be defined anywhere. Typos ----- * Line 207: holds true -> hold true

Confidence in this Review

2-Confident (read it all; understood it all reasonably well)


Reviewer 4

Summary

In this paper, the authors propose a non-convex one-pass algorithms to solve a variant of FM. The authors also prove the convergence of the proposed algorithm.

Qualitative Assessment

- The proposed algorithm seems to be promising in terms of both theoretical guarantees and computational/space complexity. - The paper is well written. - Although the main goal of this paper is on the theoretical side, the lack of any experimental results is the main issue of this paper, which limits the practical impact of this paper.

Confidence in this Review

2-Confident (read it all; understood it all reasonably well)


Reviewer 5

Summary

This paper proposed an alternating minimization approach to factorization machine and rank-one matrix sensing. The algorithm is computationally efficient and takes only one-pass of the data. The theoretical analysis provides linear convergence to the ground-truth under the assumptions that the data matrix is from normal distribution and the sample complexity is O(k^3d). This paper also gave convergence analysis in the noisy setting.

Qualitative Assessment

As far as I know, this is the first work that provides theoretical guarantees for rank-one matrix sensing with symmetric sensing matrix using alternating minimization approach. The main technique that overcomes the difficulty from the symmetric sensing matrices is a different version of RIP condition, called shifted CI-RIP, combined with a mini-batch algorithm. Therefore, I vote oral-level for novelty/originality and poster-level for potential impact. My main concern about this paper is the applicability of matrix Bernstein's inequality for the proof of Theorem 4. To apply matrix Bernstein's inequality, the authors bound \|B_i - E B_i\| with some probability on Line 409-410. However, according to my understanding about the conditions of matrix Bernstein's inequality, this term must be bounded uniformly, which means Bernstein's inequality is not applicable here since this term is bounded for every i=1,2,...,n only with some probability that is less than 1. Please see more discussion and potential solutions for this uniform boundedness assumption on Remark 5.42 and Remark 5.54 in R. Vershynin, Introduction to the non-asymptotic analysis of random matrices. arXiv:1011.3027. Therefore I vote sub-standard for technical quality. Overall, the paper is well-written except for some typos: 1. On Line 1 in Algorithm 1, two y^* -> y, X -> X^{(t)} 2. On Line 4 in Algorithm 1, the definition of \mathcal{A}(M) should be a vector rather than a scalar. 3. The equation between Line 171 and 172, \delta -> \delta_k 4. On Line 216-217, H_1 seems to be a matrix while H_2 is a scalar. Matrix and scalar are misused in many formulations, such as tr(M^* - M^{(t)}) -> tr(M^* - M^{(t)})I and H_1 - H_2 -> H_1 - H_2I 5. On Line 252, M^* -> M^*_k ========== To authors' feedback. The authors' feedback doesn't resolve my concern. The matrix Bernstein's inequality is not directly applicable if the term \|B_i - E B_i\| is not bounded uniformly, which is the case of this paper. So it is not simply a matter of union bound. However, it should be possible to resolve it by defining a new set of random variables, Z_i, such as Z_i = B_i if \|B_i\| < C; Z_i = 0 otherwise, and then applying Bernstein's inequality on Z_i. I hope the authors can deal with it formally in future versions.

Confidence in this Review

2-Confident (read it all; understood it all reasonably well)


Reviewer 6

Summary

The authors present an algorithm for efficiently recovering the parameters of “Factorization Machines”. This problem seems like a very natural extension of the rank-one matrix sensing problem. Essentially you are given samples y = x^Tw + x^TMx where x is a random gaussian vector, w is an unknown parameter vector, and M is a low-rank (or nearly low-rank) matrix. The goal is to recover w and M efficiently with as few samples as possible. For the well studied case when w is 0, existing algorithms are based on nuclear norm convex optimization methods. Unfortunately these methods are not easily extended for general vectors w, so the authors seek an alternative approach. In particular, they apply a “batch alternating minimization” algorithm that uses fresh samples at each minimization step. Batching is required because they require an l2 RIP property of the measurements which needs to hold for all low-rank matrices that come up in the alternating minimization algorithm. Unfortunately, this property cannot be made to hold with high probability over *all* rank k matrices. However, it will hold for a single matrix with good probability, so by drawing fresh samples at every step they can ensure that it hold for all intermediate low rank matrices. They give full theoretical guarantees for the algorithm, which are simple and basically optimal in the noiseless case but get more complex for the noisy case. I’m not familiar enough with the area to know whether or not the guarantees given are strong or not. As expected, and as should be necessary, the noisy case requires spectral gap dependencies since the goal is to exactly recover the top k singular vectors of the underlying matrix M.

Qualitative Assessment

Besides minor grammatical and language issues, the paper is pretty well written. The authors do a good job explaining the problem and the main difficulties that have prevented its solution. It seems like a well motivated problem, although I’m not an expert in the area. I had some questions and minor comments: To me at least, it seems very natural to use a “for each” RIP condition when a “for all” one is unobtainable. I was surprised to see the technique highlighted so prominently. Was the challenge in coming up with exactly the right guarantee and then proving it holds for random Gaussian matrices? “Compared with SVD which requires O(kd^2)” — There aren’t really any rank k SVD algorithm that run in time O(kd^2). Classical methods will require O(d^3) time. Iterative methods (i.e. power method) do require O(kd^2) time per iteration, but then this runtime is multiplied by an iteration count that depends on the accuracy desired. “For simplicity, we assume that sigma_i >= sigma_{i+1}”. I don’t really understand this statement — is this actually an assumption on M or are you just not arranging eigenvalues in the standard order but rather by absolute value? In these problems is M usually assumed to be PSD?

Confidence in this Review

1-Less confident (might not have understood significant parts)